# Characteristics of Mango Leaf Photosynthetic Inhibition by Enhanced UV-B Radiation

**Hong Wang [1], Yujian Guo [2], Jianjun Zhu [1], Kun Yue [2] and Kaibing Zhou [2,*]**

[1] Wenzhou Vocational College of Science and Technology, Wenzhou 325006, China; litchi202111@163.com (H.W.); zjj2011655@163.com (J.Z.)
[2] Engineering Research Center of Tropical Crop New Variety Breeding, Ministry of Education, Hainan University, Haikou 570228, China; yujianguo007@163.com (Y.G.); 990952@hainanu.edu.cn (K.Y.)
[*] Correspondence: zkb@hainanu.edu.cn

**Abstract:** To investigate the photosynthetic change characteristics of mango leaves under enhanced UV-B radiation, adult 'Tainong No. 1' mango (*Mangifera indica*) trees were treated (N = nine individuals) with simulated enhanced UV-B radiation [24 and 96 kJ/(m$^2$·d)] in the field, and the photochemical reactions, activities of key enzymes in carbon assimilation, and the expression of genes were observed. The results showed that compared with the control, there was a decrease in tree yield, soluble sugar, sugar–acid ratio, and vitamin C of the fruits under the 96 kJ/(m$^2$·d) treatment, while no significant changes were observed under 24 kJ/(m$^2$·d). After 20 or 40 days, the leaves' net photosynthetic rate (Pn), stomatal conductance (Sc), transpiration rate (Tr), intercellular $CO_2$ concentration (Ci), and chlorophyll *a/b* under exposure to 96 kJ/(m$^2$·d) of UV-B were significantly lower than in the control, whereas chlorophyll *a*, chlorophyll *b*, carotenoids, Hill reaction activity, photochemical quenching coefficient (qP), and Rubisco activities were significantly higher. In contrast, the Hill activity and Rubisco activity under 24 kJ/(m$^2$·d) were significantly higher than the control, and increased by 350% and 30.8%, respectively, while Pn, Sc, Tr, Ci, and the content of photosynthetic pigments were similar to the control. The expression of gene coding the Rubisco big subunit (*rbcL*) was inhibited by the 96 kJ/(m$^2$·d) treatment. We conclude that stomatal limitation was directly induced by 96 kJ/(m$^2$·d), resulting in the inhibition of photosynthesis and the reduction in yield and deterioration of the quality of mango.

**Keywords:** *Mangifera indica*; photosynthesis; stomatal limitation; enhanced UV-B radiation





## 1. Introduction

Ultraviolet radiation zone B (UV-B, wavelength 280–320 nm) is largely absorbed by the ozone layer, with only a small amount reaching the ground. Chlorofluorocarbon contamination in the stratosphere has resulted in a decrease in ozone, which in turn has led to an increase in UV-B radiation [1,2], which is referred to as enhanced UV-B radiation. Without proper control of pollutants, ozone in the stratosphere will continue to decline. It is estimated that the annual increase rate of enhanced UV-B radiation in the Northern Hemisphere will reach 14% and that in the Southern Hemisphere will reach 40% [3]. The damage, in turn, has raised concerns about the potential harm of enhanced UV-B radiation.

Light is a necessary energy condition for plant photosynthesis, and photosynthetic systems have become targets for enhancing the effects of UV-B radiation stress. Enhanced UV-B radiation can negatively affect photosynthetic machinery, leading to loss of thylakoid membrane integrity, damage to the photosystem II (PSII), reduced $CO_2$ assimilation, and reduced oxygen release as well as other effects [4–7]. In addition, UV-B radiation has been considered historically as an environmental stress factor, and the specific effects on plants include plant dwarfing [8,9], biomass reduction [10,11], photosynthetic rate reduction [12,13], DNA damage and repair [14,15], etc.

The ultraviolet radiation is strong in Hainan and other tropical low latitudes, so the effect on tropical perennial fruit trees is particularly prominent. Photosynthesis is the material basis for the formation of crop yields, so improving the photosynthetic efficiency of fruit trees has a decisive influence on the yield and quality of fruit trees. It is necessary to carry out a prospective study on the physiological causes of the effects of enhanced UV-B radiation on the cultivation performance and photosynthesis of mango adult trees in the future. Therefore, the effects of enhanced UV-B radiation on the cultivation performance and photosynthesis of 'Tainong No. 1' mango were studied by enhanced UV-B radiation in the field artificial simulation room. The physiological mechanism of the effect of enhanced UV-B radiation on photosynthesis in mango leaves is summarized.

## 2. Materials and Methods

### 2.1. Plant Material and Growth Conditions

The experiments were conducted in Juntian Village, Yingzhou Town, Lingshui County, Hainan Province, China. This site has a tropical monsoon and marine climate with abundant sunshine and rainfall. The annual mean temperature, sunshine duration, and rainfall are 25.4 °C, 2261.6 h, and 1717.9 mm, respectively. The soil is a fertile latosol. Nine 10-year-old uniform trees of 'Tainong No.1' mango were selected for sampling.

### 2.2. UV-B Radiation Experiments

Using natural light as the control (CK), two treatment levels of 24 kJ/(m$^2$·d) (low dose) and 96 kJ/(m$^2$·d) (high dose) were used, which is approximately equivalent to a 15% increase in UV-B radiation in normal environments. Since 12 February 2018, UV-B lamp tubes have been suspended at the center of the top of the test trees (40 W, radiation range of 280–320 nm, and radiation dose of 24 kJ/(m$^2$·d); Beijing Electric Light Source Research Institute) to simulate increased UV radiation. The outer layer of the light source was covered by a 0.08 mm cellulose acetate membrane to filter the UV-C radiation. The intensity of the different UV-B treatments was adjusted by changing the number of light sources. The distance between the lamp tube and the top of the tree was kept at 30 cm, and the growth of new shoots was controlled after flowering and fruiting in order to maintain the height throughout the treatment period. Artificially simulated enhanced UV-B radiation treatment was carried out on sunny days. The treatment time was from sunrise to sunset. The lights were turned off during cloudy and rainy weather as well as on cloudy days.

### 2.3. Sampling for Biochemical Assays

Ten mango leaves from each tree were collected every 20 d from 12 February 2018 to 8 May 2018 for biochemical analysis. The yield per plant was investigated during fruit harvesting, and five fruit samples were randomly selected from the periphery of the middle crown for quality analysis. Leaf and fruit samples were collected and frozen in liquid nitrogen in time and stored at −80 °C until use.

### 2.4. Indicators and Methods

Field investigation of plant yield. The contents of soluble sugar, leaf chlorophyll (Chl), and vitamin C (Vc) in the fruits were determined according to the method of Li He-sheng [16]. The titratable acid can be determined by the NaOH standard solution titration method [17], and the sugar–acid ratio of the fruits was converted. The photosynthetic rate (Pn), stomatal conductance (Sc), transpiration rate (Tr), and intercellular $CO_2$ concentration (Ci) were measured using a LI-6400 portable photosynthetic system (LI-COR Biotechnology Company, Lincoln, NE, USA) in the morning (between 09:30 and 11:30). Hill reaction activity was measured according to Zhang Shuqiu's method, where the functional leaves were cut into pieces, ground with liquid nitrogen, and then the chloroplasts were extracted, added with an appropriate amount of potassium ferricyanide, illuminated, the OD value of the reaction solution was measured, and the standard curve checked to obtain the chlorophyll per milligram per unit time. The amount of oxygen released [18].

The fluorescence quenching coefficient of Chl was measured by a MINI-PAM portable pulse modulation fluorometer (Zealquest Scientific Technology Co. Ltd. Shanghai, China). Ribulose bisphosphate carboxylase oxygenase (Rubisco) activity was determined by a kit (Kote Biotechnology Co. Ltd., Jiangsu, China). A kit was used to extract RNA from the mango leaves (GENEDENOVO Company, Guangzhou), following which the cDNA was reverse-transcribed (Tiangen, Beijing, China). Primers were designed based on the existing *rbcL* gene sequence conserved region and the internal reference gene *actin:* RBCL1 (5′-CGTTACAAAGGACGATGCTACAA-3′), RBCL2(5′-GAACCCAAATACATTACCCACAA-3′), aF (5′-GCTGAGAGATTCCGATGCCC-3′), aR (5′-TGATGGAGTTGTAGGTGGTCT-3′). The coding region of the *rbcL* gene was amplified, and quantitative real-time (qRT)-PCR was performed to detect the relative expression of the gene.

### 2.5. Data Analysis

SAS 9.0 (Statistical Analysis System, Cary, NC, USA) software was used for analysis of variance (ANOVA), followed by Duncan's multiple range tests for the evaluation of significant differences among treatment means. Least significant differences were calculated for cases where treatment means were significantly different at $p = 0.05$.

## 3. Results

### 3.1. Yield and Quality

UV-B had no significant impact on the yield, soluble sugar, titratable acid, and sugar–acid ratio, whereas the vitamin C content significantly increased in fruits exposed to 24 kJ/(m$^2$·d) (Table 1). In contrast, exposure to 96 kJ/(m$^2$·d) of UV-B significantly reduced the yield, soluble sugar, and sugar–acid ratio, whereas the titratable acid and vitamin C content significantly increased in fruits under this treatment (Table 1). Generally, exposure to 96 kJ/(m$^2$·d) of UV-B reduced the yield and quality compared with non-irradiated plants, while treatment with 24 kJ/(m$^2$·d) of UV-B had no significant effect (Table 1).

**Table 1.** The effects of enhanced UV-B radiation on the yield of single trees and the quality of mango fruits.

| Treatment | Yield/Kg | Soluble Sugar/% | Titratable Acid/% | Sugar–Acid Ratio | Vitamin C/(mg/100 g) |
|---|---|---|---|---|---|
| CK | (26.20 ± 1.66) [a] | (13.85 ± 0.57) [a] | (0.49 ± 0.03) [b] | (28.27 ± 0.82) [a] | (131.7 ± 0.23) [b] |
| 24 kJ/(m$^2$·d) | (31.42 ± 2.73) [a] | (14.39 ± 0.56) [a] | (0.58 ± 0.08) [b] | (24.81 ± 1.05) [a] | (182.3 ± 0.60) [a] |
| 96 kJ/(m$^2$·d) | (17.32 ± 2.02) [b] | (11.16 ± 0.79) [b] | (0.83 ± 0.05) [a] | (13.45 ± 1.22) [b] | (168.8 ± 1.12) [a] |

Note: The values followed by different small letters differ significantly at $p < 0.05$.

### 3.2. Pn, Sc, Tr, and Ci

The Pn and under 24 kJ/(m$^2$·d) UV-B were significantly lower than in the control by day 20, and no significant difference from the control was detected thereafter (Figure 1A,C). Ci under 24 kJ/(m$^2$·d) of UV-B was significantly higher than the control only on day 20, whereas Tr was significantly lower than the control from d 20 to 80 (Figure 1B,D). Conversely, exposure to 96 kJ/(m$^2$·d) UV-B significantly reduced Pn, Sc, Tr, and Ci from day 20 to 80 (Figure 1A–D).

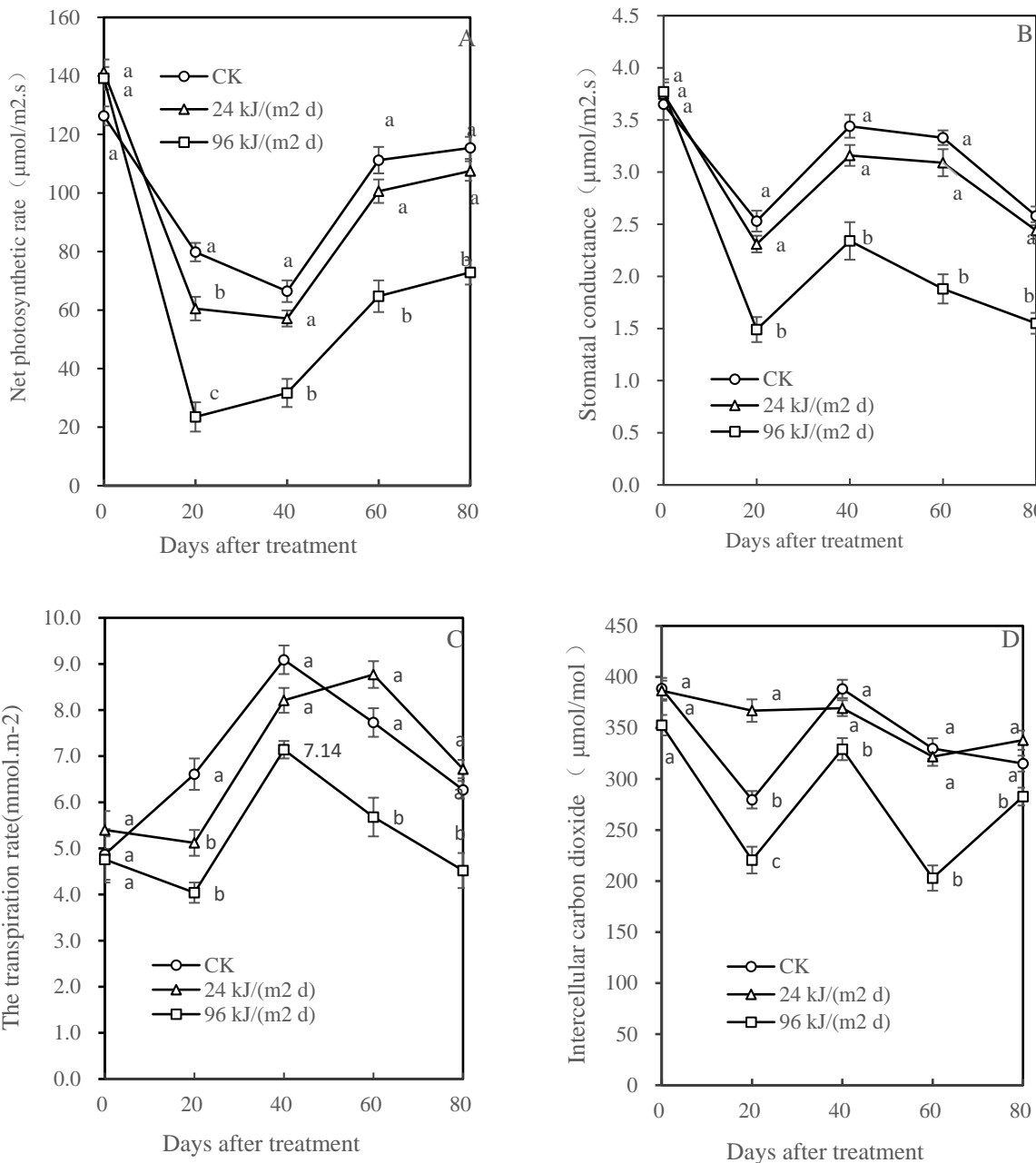

**Figure 1.** Net photosynthetic rate (Pn) (**A**), stomatal conductance (Sc) (**B**), transpiration rate (Tr) (**C**) and intercellular $CO_2$ concentration (Ci) (**D**) of mango leaves exposed to 24 kJ/(m$^2$·d) and 96 kJ/(m$^2$·d) of UV-B for 20 and 80 d. Data are the means (±standard errors, SEs) of nine replicates per treatment, respectively. The same letters on the points indicate no significant difference at $p < 0.05$.

It is evident that exposure to 96 kJ/(m$^2$·d) UV-B decreased the leaf net photosynthetic rate by decreasing Sc and Ci, thus inhibiting leaf transpiration, which is an important contributor to the decline in plant yield and the deterioration in fruit flavor quality.

### 3.3. Chl a, Chl b, Chl a/b, and Carotenoids

The Chl content increased significantly only on day 40 under 24 kJ/(m$^2$·d) UV-B irradiance when compared with the control (Figure 2A). However, there was no significant difference in Chl *b*, Chl *a/b*, and carotenoid contents between the 24 kJ/(m$^2$·d) treatment and the control at all time points (Figure 2B–D).

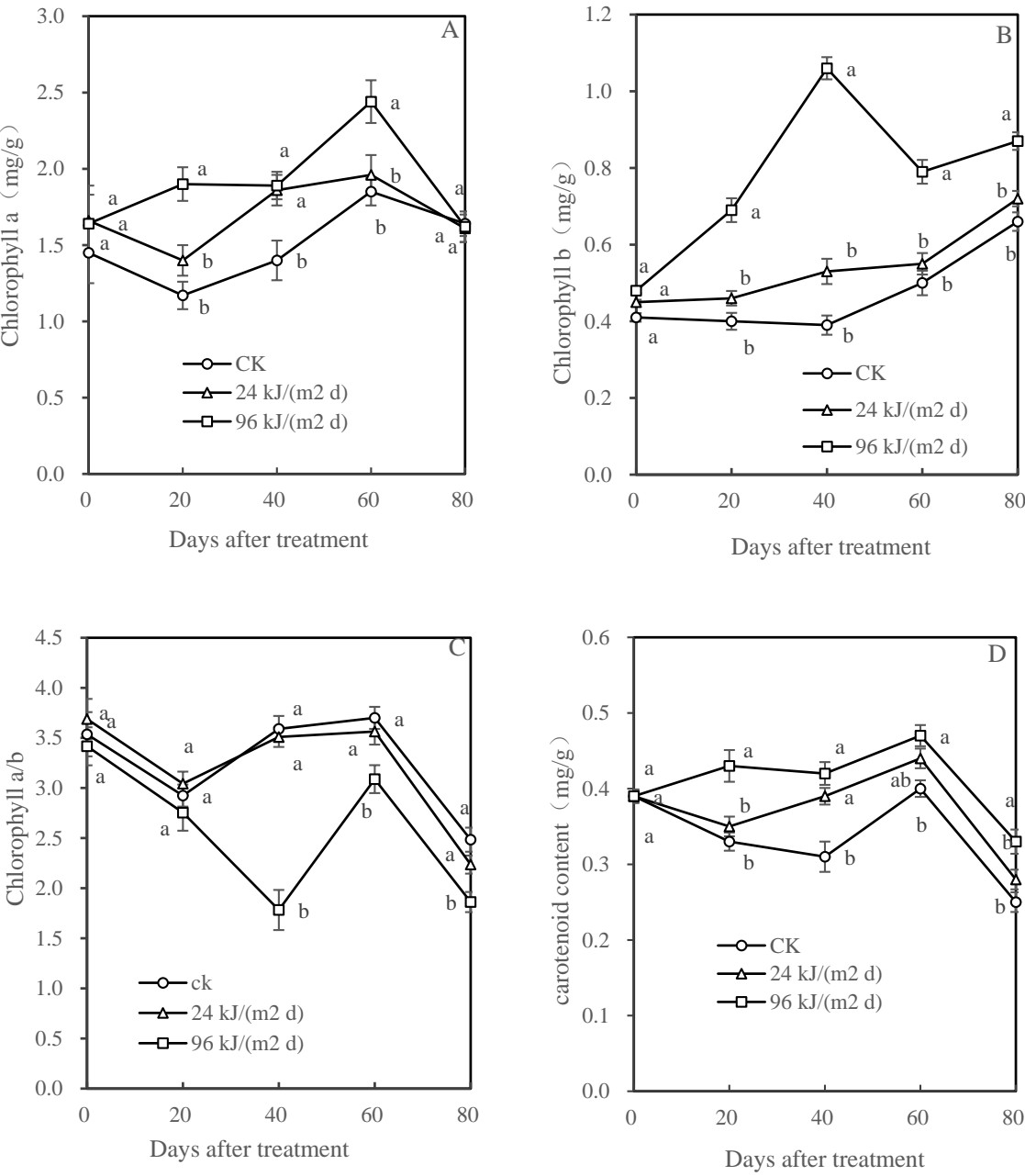

**Figure 2.** Chlorophyll a (**A**), chlorophyll b (**B**), chlorophyll a/b (**C**), and carotenoid (**D**) of mango leaves exposed to 24 kJ/(m²·d) and 96 kJ/(m²·d) of UV-B for 20 and 80 d. Data are the means (±standard errors, SEs) of nine replicates per treatment, respectively. The same letters on the points indicate no significant difference at $p < 0.05$.

A statistically significant increase after 20 day was observed in Chl *b* and carotenoids at 96 kJ/(m²·d) UV-B intensity, while in Chl *a/b*, this increase was exhibited after 40 days (Figure 2B–D). The Chl *a* content under 96 kJ/(m²·d) of UV-B was significantly higher than the control from days 20 to 60 (Figure 2A).

The results showed that enhanced UV-B radiation treatment of 96 kJ/(m²·d) increased the contents of Chl *a,* Chl *b*, and carotenoids, but significantly decreased the Chl *a/b* value, which is generally considered to be more reflective of leaf photosynthetic capacity. The higher the Chl *a/b* value within a certain range, the stronger the photosynthetic capacity; therefore, the high-dose treatment showed a tendency to inhibit leaf photosynthetic capacity.

### 3.4. Effects of UV-B Radiation on Light Reaction in Mango Leaves
Hill Activity and qP

There was no significant difference in Hill activity in the mango leaves between the two UV-B radiation treatments and the control before 20 day. However, the UV-B radiation of both 24 kJ/(m$^2$·d) and 96 kJ/(m$^2$·d) resulted in higher Hill activity in the leaves than in the control from days 40 to 80 (Figure 3A).

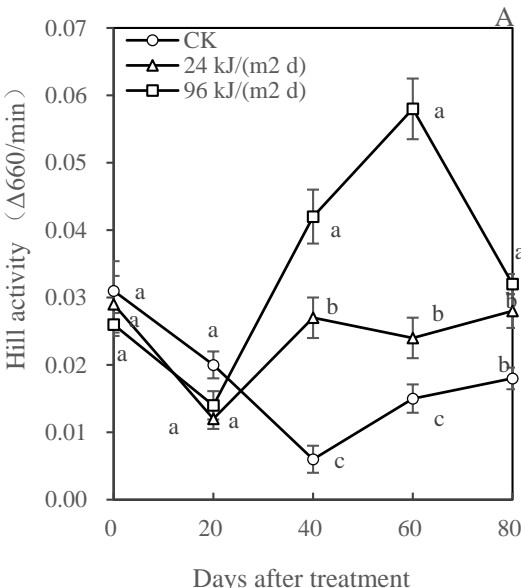
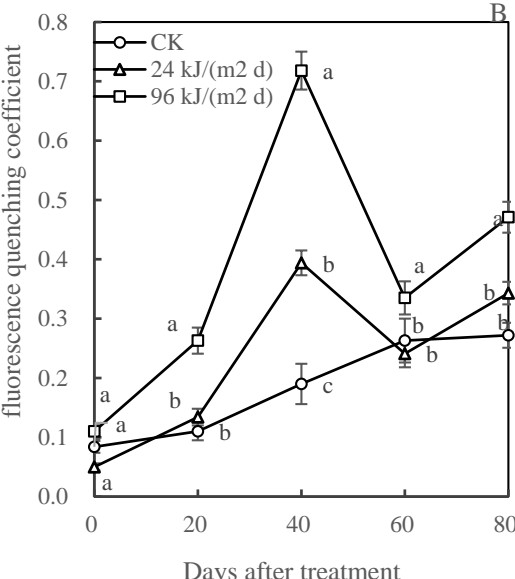

**Figure 3.** Hill activity (**A**) and photochemical quenching coefficient (qP) (**B**) of mango leaves exposed to 24 kJ/(m$^2$·d) and 96 kJ/(m$^2$·d) of UV-B for 20 and 80 d. Data are the means (±standard errors, SEs) of nine replicates per treatment, respectively. The same letters on the points indicate no significant difference at *p* < 0.05.

The qP of 24 kJ/(m$^2$·d) and 96 kJ/(m$^2$·d) UV-B radiation treatment increased steadily, peaking after 40 d and stabilizing thereafter (Figure 3B). The difference in qP between the 96 kJ/(m$^2$·d) treatment and the control was significant from days 20 to 80, whereas the 24 kJ/(m$^2$·d) treatment was only significant on day 20 (Figure 3B).

From our above data analysis, it is evident that both 24 kJ/(m$^2$·d) and 96 kJ/(m$^2$·d) UV-B radiation promoted the photolysis of water and the formation of an assimilation force, which in turn promoted photoreactions. In addition, the 96 kJ/(m$^2$·d) treatment may improve the efficiency of leaf light energy transfer and promote light reactions.

### 3.5. Effects of UV-B Radiation on the Carbon-Fixation Reaction in Mango Leaves
Rubisco Activity and Gene Expression Analysis

Both the 24 kJ/(m$^2$·d) and 96 kJ/(m$^2$·d) UV-B radiation treatments showed an increasing trend in Rubisco activity and were significantly higher than the control (Figure 4A).

The relative expression of the *rbcL* encoding gene in the 24 kJ/(m$^2$·d) treatment was significantly higher than the control after 60 d, but was significantly lower than the control or did not differ from the control at other time points. The relative expression of *rbcL* in the 96 kJ/(m$^2$·d) treatment was significantly lower than in the control from days 20 to 80 (Figure 4B).

The results showed that both UV-B radiation treatments could improve CO$_2$ fixation by increasing the Rubisco activity in the leaves and improving the carbon-fixation reaction. In addition, the expression of *rbcL* also decreased in the 96 kJ/(m$^2$·d) treatment.

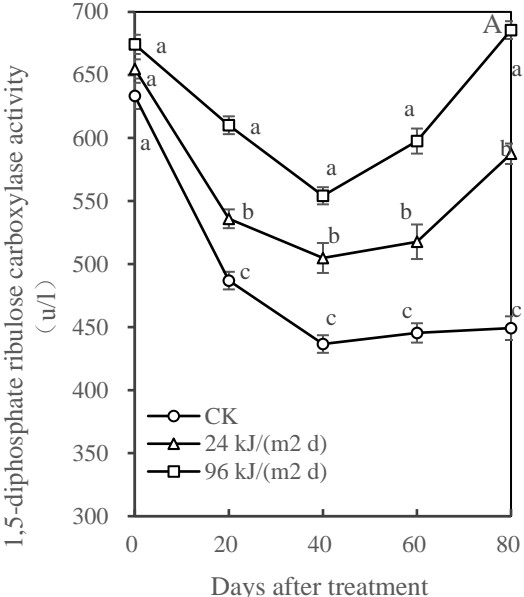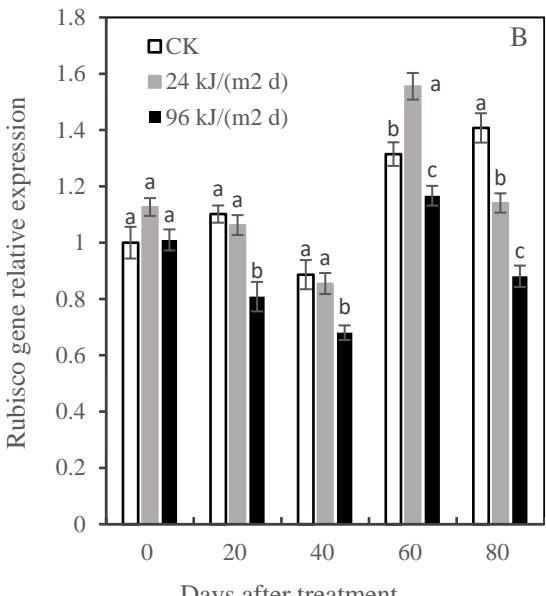

**Figure 4.** Rubiscol activity (**A**) and Rubisco gene relative expression (**B**) of mango leaves exposed to 24 kJ/(m$^2$·d) and 96 kJ/(m$^2$·d) of UV-B for 20 and 80 d. Data are the means (±standard errors, SEs) of nine replicates per treatment, respectively. The same letters on the points indicate no significant difference at $p < 0.05$.

## 4. Discussion

### 4.1. Stomatal Limitation under Enhanced UV-B Radiation Inhibits Photosynthesis in Mango Trees

Enhanced UV-B radiation decreases the leaf transpiration rate, stomatal conductance, and stomatal resistance, thus reducing intercellular $CO_2$ concentration, which, in turn, affects $CO_2$ assimilation efficiency, resulting in a decrease in crop photosynthesis and productivity by causing stomatal restriction and inhibiting leaf photosynthesis [19–21]. In our previous studies, the 'Jinhuang' mango orchard in Baoyou Town, Ledong County, China as well as the 'Tainong No. 1' mango in an orchard in Yingzhou Town, Lingshui County, China, in 2016 and 2017 were used as test materials [12,13]. The results showed that high-intensity UV-B radiation could cause stomatal limitation in the leaves of adult mango trees, corroborating the results of the above previous studies. In this paper, an experimental study was carried out on the adult mango tree 'Tainong No. 1' in another mango orchard in Yingzhou Town, Lingshui County in 2018, and the results were consistent with the above findings. Based on the results of these experiments, which were conducted across several years using different mango varieties, it can be concluded that high-intensity UV-B radiation can cause leaf stomatal limitation, which is one of the mechanisms by which leaf photosynthesis is inhibited in adult mango trees.

### 4.2. Non-Stomatal Limitation Caused by Enhanced UV-B Radiation Inhibits Photosynthesis and Causes Fruit Damage in Mango

Kataria S reviewed the harmful effects of UV-B radiation on plant photosynthesis and photosynthetic productivity. The literature revealed that UV-B radiation inflicts damage to the photosynthetic apparatus of green plants at multiple sites. The sites of damage include oxygen evolving complex, D1/D2 reaction center proteins, and other components on the donor and acceptor sides of PS II [22]. The effects of enhanced UV-B radiation on plant photosynthetic organs and photosynthetic processes are referred to as direct effects [23]. Adverse direct effects can inhibit leaf photosynthesis, resulting in non-stomatal limitation. The non-stomatal limitation associated with increased UV-B radiation on plant photosynthesis can be summarized in the following three points. First, UV-B radiation decreases Chl and carotenoid content, destroys the PSII reaction center, inhibits the electron transport of PSII, and decreases Hill reaction activity and the chlorophyll fluorescence quenching

coefficient [24,25]. Second, the activities of two important enzymes in photosynthetic carbon assimilation [Rubisco and phosphoenolpyruvate carboxylase (PEPC)] are decreased, leading to reduced $CO_2$ carboxylation capacity [26,27]. Third, photosynthetic genes are downregulated such as *Chl* genes, *psbA*, nuclear genes, and the Rubisco large and small subunit genes [28]. With the exception of the downregulation of the *rbcL* coding gene of the Rubisco large subunit in the leaves treated with 96 kJ/(m$^2$·d), our results were generally contrary to the above previous results. However, our findings demonstrated that the Chl *a/b* value of the leaves decreased significantly under the 96 kJ/(m$^2$·d) treatment. As the Chl *a/b* value is positively correlated with the stacking degree of the thylakoids in the chloroplasts and the photosynthetic capacity of the leaves [29], this indicates that the stacking structure of the chloroplast thylakoid was damaged by high-intensity UV-B radiation, resulting in inhibited leaf photosynthesis. This may increase the non-stomatal limitation of mango leaves under high UV-B radiation. The present study is the first to report this conclusion. The morphological and structural changes of the chloroplasts induced by high-intensity UV-B radiation are based on the changes in their chemical components such as photosynthetic pigments, pigment proteins, photosynthetic chain electron transfer proteins, membrane proteins, and membrane lipids. The damage to chloroplast morphology and structure can be confirmed by electron microscope observation of the leaves treated with enhanced UV-B radiation.

The content of carotenoids in the isolated leaves of mango and *Caryopteris mongholica* increased after UV-B radiation treatment, which may be a protective mechanism for absorbing UV-B radiation and dispersion stress, and also enhances the scavenging of reactive oxygen species [30,31]. The findings of this paper are consistent with these results. In addition, it has also been reported that enhanced UV-B radiation stimulates an increase in photosynthetic pigment contents in the leaves [31,32]. This may be caused by the increased leaf thickness, and the increase in leaf thickness may reduce the damage of the UV-B radiation to the leaf cells [33]. It has also been reported that rice treated long-term with UV-B radiation did not exhibit any changes in the short-term. In particular, the efficiency of absorption and the transmission and conversion of light energy were improved once the rice had adapted to the stress conditions [33,34]. These results are consistent with the results of this experiment, but are contrary to the typical characteristics of photosynthetic non-stomatal limitation.

The changes in chlorophyll fluorescence and chloroplast photochemical activity in different plant species have different characteristics. The increase and decrease in photosynthetic pigment content may be insufficient to explain whether non-stomatal limitation had inhibited photosynthesis in the leaves. The detection of photochemical activity in vitro may not fully reflect the fact that the photosynthetic system plays its physiological function in vivo. Different photosynthetic elements must coordinate under a reasonable structured system in order to ensure the normal physiological functioning of the photosynthetic system. In summary, high UV-B radiation inhibited photosynthesis by causing both stomatal limitation and non-stomatal limitation, resulting in a decrease in plant yield and deterioration in fruit flavor quality.

The results showed that the 96 kJ/(m$^2$·d) treatment increased Rubisco activity and downregulated the expression of *rbcL* in the leaves, indicating that the change in Rubisco activity may not be directly related to the change in gene expression efficiency. In fact, previous reports have indicated that there is no significant correlation between the activities of $\alpha$-amylase, cysteine protease, and glutathione reductase, or the expression level of coding genes [35–37]. It is speculated that complex changes may have occurred in the process of post-transcriptional translation regulation of the Rubisco gene and protein processing after translation, which may also be related to the changes in many factors affecting enzyme activity. The specific mechanism requires further investigation.

## 5. Conclusions

High-dose [96 kJ/(m$^2$·d)] treatment could directly cause leaf stomatal limitation to inhibit leaf photosynthesis, which ultimately led to a decrease in the yield of the adult trees and a decline in the fruit flavor quality. The low dose (24 kJ/(m$^2$·d)) treatment had no significant effect on the photosynthesis, plant yield, and fruit quality of the adult trees. This study informs on fruit tree photosynthesis and anti-light stress physiology, and can guide the development of reasonable cultivation techniques for improving the adaptation of mango enhanced UV-B radiation. We will see that grain stacking is based on protein–protein interactions, especially involving photosystem II and light harvesting complexes. Thus, gene expression (stabilizing the transcription factor and the protein import system at the chloroplast envelope) may be a feature worthy of attention.

**Author Contributions:** Conceptualization, K.Z.; methodology, H.W. and K.Z.; software, H.W., K.Y. and Y.G., validation, H.W., K.Y. and Y.G.; formal analysis, H.W.; investigation, H.W., K.Y. and Y.G., resources, K.Z. and J.Z.; data curation, H.W. and K.Z.; writing—original draft preparation, H.W.; writing—review and editing, H.W., K.Z. and J.Z.; visualization, H.W. and Y.G.; supervision, K.Z.; project administration, K.Z.; funding acquisition, K.Z. and J.Z. All authors have read and agreed to the published version of the manuscript.

**Funding:** This work was supported by the National Natural Science Foundation of China (Project No. 31460498) and Open Research Subject of Wenzhou Key Laboratory of Horticultural Plant Breeding (Project No. ZD202003).

**Institutional Review Board Statement:** Not applicable.

**Informed Consent Statement:** Not applicable.

**Data Availability Statement:** Not applicable.

**Acknowledgments:** We thank LetPub for the linguistic assistance during the preparation of this manuscript.

**Conflicts of Interest:** The authors declare no conflict of interest.

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
