# Peer review of "Characteristics of Mango Leaf Photosynthetic Inhibition by Enhanced UV-B Radiation"

_horticulturae, doi:10.3390/horticulturae7120557_

Round 1

Reviewer 1 Report

The present study evaluated the impact of UV-B radiations on leaf photosynthesis inhibition of mango. The manuscript present some interesting results but need improvement before consider for publication in Horticulturae. The major concerns is regarding the material and method section as detail of key experimental procedures and design are missing (see comments annotated on pdf file)

Author Response

Respected reviewer,

     Good lucky for you!  The first author had finished the modification, and she had answered the guidances for the modifications of this paper.  I have send her reply to you for details in the attachment. Thank you very much!

Reviewer 2 Report

Dear authors, thank you for this well structured manuscript on „Characteristics of Mango Leaf Photosynthetic Inhibition by Enhanced UV-B Radiation“. You are facing the problem to serve two groups of colleagues interested in different aspects. The title will attract colleagues interested in basic research (the biochemistry of photosynthesis under UV stress). But you also are addressing colleagues interested in breeding mangos for improved UV resistance and fruit quality. It will be not easy to satisfy both of them.
In your discussion chapter you are focussing on symptoms while you are sort of ignoring earlier publications describing UV effects on photosynthesis in more detail. You might take a closer look at the review of Sunita Kataria, Anjana Jajoo, Kadur N. Guruprasad (2014) Impact of increasing Ultraviolet-B (UV-B) radiation on photosynthetic processes. Journal of Photochemistry and Photobiology B: Biology 137 (2014) 55–66. These authors are providing a more balanced view, though they are not telling details on photosynthesis.
In your discussion you are focussing on the observed differences between high and low UV-B dose. Due to your experimental set up, you have to do so. But in the summary you might mention effects of short and long time exposure as well. You have mentioned long term effects on leaf anatomy, but you also might mention in this context the decrease of UV-B intensity on its path through the leaf tissue. 
Breeders will be a bit unhappy with the results, because you can’t identify a single gene to be modified. You argue that most of the observed effects can be explained by UV-B effects on stomata closure. Well, this is correct. But in the literature this is classified a secondary effect. - You also have identified effects on gene expression (rubisco protein, for instance). If you will examine literature on primary reactions in photosynthesis and the chloroplast anatomy, you will see that grana stacking is based on protein-protein interactions especially involving photosystem II and light harvesting complexes. Thus, gene expression (stabilizing the transcription factor and the protein import system at the chloroplast envelope) may be a trait to look at. - I disagree with your suggestion made in lines 257-259: Relevant literature is available, this problem has been solved 20-30 years ago.

Author Response

(The authors gave the same response as above.)

Reviewer 3 Report

Main comments:

The content of the manuscript is interesting and novel in which the characteristics of inhibiting photosynthesis of mango leaves by enhanced UV-B radiation are reported. The research on the quality of mango fruit and gene expansion under these radiation conditions is also very valuable.

Title: my suggestion that the title should include information about the effect on the yield and quality of the fruit.

Material and methods:

Line 88: I suggest the authors describe these methods briefly because

the cited literature is less known and available.

Line 93: Zhang Shuqiu et al. [18]. This publication is also little known, and it is not included in the reference of this manuscript.

The size of the font should also be equal.

Results: Figures: Figure 1, 2, 3 (Figuare 1).

Values ​​with descriptions overlap on the vertical axis.

Reference:

Line 280: it should be in italic: Vaccinium corymbosum

this also applies to the other publications listed.

Author Response

(The authors gave the same response as above.)
